# A Discrete-Finite Element Analysis Model Based on Almen Intensity Test for Evaluation of Real Shot Peening Residual Stress

**DOI:** 10.3390/ma16155472

**Published:** 2023-08-04

**Authors:** Chengan Wang, Yujin Park, Taehyung Kim

**Affiliations:** 1Graduate School, Department of Mechanical and Aeronautical System Engineering, Cheongju University, 298 Daeseong-ro, Cheongju-si 28503, Republic of Korea; wca1782618488@163.com (C.W.); yudng@cju.ac.kr (Y.P.); 2Department of Aeronautical and Mechanical Engineering, Cheongju University, 298 Daeseong-ro, Cheongju-si 28503, Republic of Korea

**Keywords:** shot peening, residual stress, discrete element analysis, finite element analysis, Almen intensity, X-ray diffraction

## Abstract

In this study, a combined discrete-finite element model based on the Almen intensity measurement test was proposed to evaluate the real shot peening residual stress. The discrete element analysis was utilized to simulate the random behavior of numerous shot balls, while the finite element analysis was employed to quantitatively predict the residual stress induced by shot peening. Moreover, the Almen intensity, an essential factor in the actual shot peening process, was taken into account. Initially, an Almen strip analysis model was established, and the multi-random impact analysis was performed to validate the good agreement between the analytical Almen curve and experimental Almen curve. Subsequently, the unit cell discrete-finite element analysis model was expanded for predicting the peening residual stress, incorporating the Almen intensity. The analysis results showed a significant correlation between the predicted peening residual stress and the XRD (X-ray diffraction) experimental residual stress. Therefore, it was confirmed that the proposed discrete-finite element random impact analysis model in this study could serve as an effective analytical technique capable of substituting for the actual shot peening process.

## 1. Introduction

Shot peening technology is widely used in various industrial fields such as aircraft, industrial power equipment components, railway vehicles, and automotive durable parts [1,2,3]. Shot peening is a surface treatment technique that induces compressive residual stress by subjecting the surface of a component or metal material to the continuous multiple impacts of numerous small shot particles, causing plastic deformation. Therefore, when shot peening is applied to the surface of a mechanical component subjected to repetitive tensile stress, the compressive residual stress resists the tensile stress and extends the fatigue life of the component [4,5,6]. To quantitatively analyze the effects of such compressive residual stress, it is necessary to predict the magnitude of the surface and maximum compressive residual stress, as well as the depth of the residual stress layer. Typically, peening residual stress generated on the surface of fatigue-prone components is measured using X-ray diffraction [7,8,9]. However, this method poses potential harm to the operator due to the use of radiation. Moreover, the preparation and measurement times for the measurement vary depending on the operator’s skill level, and the measurement results are not consistent, making it difficult to quantitatively predict residual stress. For these reasons, various studies based on finite element analysis have been conducted to quantitatively predict the multi-impact shot peening residual stress.

Recently, numerical analysis studies utilizing three-dimensional random impact have been introduced to simulate the phenomenon of actual shot particles randomly impacting the material surface [10,11,12,13,14,15,16,17,18,19,20,21]. Kim et al. and Meguid et al. proposed a four-fold symmetric cell analysis model that replaces a wide peening area and allows for various impact sequences perpendicular to the four vertices of the symmetric cell [10,11]. Miao et al. proposed a hexahedral analysis model that focuses on localized areas with impact areas ranging from 1/6 to 1/25 of the total area for multi-impact impact [12]. Additionally, Kim et al. and Gangaraj et al. proposed a multi-impact analysis model in the form of a disk-type plate, taking into account the effect of peening coverage. They simulated the phenomenon of random impact by concentrating the impact of shot balls in the central region of the disk [13,14]. Xiao et al. established a finite element model with shot balls randomly impacting an overall area that is approximately 2.7 times the length of a single shot ball’s diameter [15]. However, these previous studies all had in common that the positions of the shot balls were pre-determined to impact the material surface. In other words, they used random number generation functions in software such as Excel, Matlab, or Python to obtain the impact and height coordinates of the shot balls and then arranged them accordingly. Therefore, these papers have limitations in implementing the random impact positions and sequences of shot balls. Furthermore, due to the increase in analysis time caused by the deformation of shot balls upon impact with the material, the peening area and the quantity of shot balls are often limited in most cases. Therefore, recently, a new analysis technique combining discrete element analysis and finite element analysis has been introduced to simulate actual random phenomena [22,23,24,25,26,27,28]. Tu et al. first obtained the random velocities and impact positions of shot balls via discrete element analysis and then transferred them to dynamic finite element analysis to implement the multi-impact phenomenon [22]. They obtained residual stress at different depths from the surface at the impact centers of the shot balls and averaged them arithmetically. However, actual peening residual stress measured by X-ray diffraction exhibits variations in stress not only at the impact center but also in the surrounding area and the entire surface. Therefore, analytically evaluated residual stress based solely on the impact center differs from XRD experimental residual stress. Bhuvaraghan et al. performed single-impact finite element analysis to obtain pressure data from the indentation generated on the material surface [23]. They incorporated this pressure into a multi-shot random impact discrete element analysis to obtain equivalent pressure exerted on the material at the impact locations, which were also randomly obtained. Then, they obtained the peening residual stress from a finite element analysis model that incorporated only the equivalent pressure and randomly obtained impact positions instead of using a multi-shot model. However, these studies did not include experimental verification to establish the validity of the analytical techniques. Additionally, they did not consider the essential factor of peening intensity, which is crucial in predicting the peening residual stress of actual fatigue-prone components. Edward et al. first performed dynamic single-shot impact finite element analysis to calculate the restoration coefficient, which was then incorporated into the discrete element analysis [24]. After the discrete element analysis, they obtained the contact load exerted by the shot ball on the material surface and incorporated it back into the finite element analysis model to simulate the peening phenomenon even in the absence of contact between the shot ball and the material. However, they did not simulate the multi-shot impact phenomenon based on actual peening processes, and process variables such as Almen intensity and peening coverage were not considered. Furthermore, experimental residual stress that could confirm the validity of the analytical residual stress was excluded from their study. Indeed, the recent studies mentioned earlier have incorporated advanced analytical techniques in simulating actual peening phenomena. However, it is true that they have not considered the crucial factor of Almen intensity, which represents the peening intensity and encompasses variables such as the impact velocity and quantity of shot balls. Almen intensity is a key process parameter that must be taken into account during the peening process. Fatigue-prone components have diverse fatigue characteristics depending on their usage and operating environment, so it is important to apply the optimal peening intensity [29,30,31,32,33]. Therefore, the multi-random impact peening analysis model in this study, which considers Almen intensity, stands out from the existing analytical models. In this study, a new analysis technique is proposed to evaluate the real shot peening residual stress using the discrete-finite element analysis model based on the Almen intensity measurement test. Here, the discrete element analysis is used for the random behavior of numerous shot balls, and the impact load is obtained when the shot balls impact the surface of the model, while finite element analysis is used for the deformation behavior of the model, and residual stress is indicated. Ultimately, this technique will be verified by comparing the analytical residual stress with the X-ray diffraction (XRD). This approach can be effectively utilized as a valuable analytical tool for exploring optimal shot peening conditions in various durability components.

## 2. Almen Intensity Simulation and Experimental Verification

### 2.1. Almen Intensity by DE-FE Analysis

In this section, an Almen strip analysis model was developed using discrete-finite element analysis to obtain the Almen intensity corresponding to the actual experimental results and assess its suitability. In shot peening processes, the Almen intensity is a crucial factor that must be taken into account. It combines various variables such as peening time, peening coverage, shot diameter, shot velocity, and impact angle into a single parameter that represents the Almen intensity [29,30,31,32,33]. Therefore, even when simulating the actual shot peening phenomenon, a random multi-impact analysis based on the Almen intensity must be performed. To secure the Almen intensity, an Almen strip analysis model was established to obtain the analytical Almen intensity. The Almen strip analysis model targeted an A-type strip with dimensions of 76 mm in length, 19 mm in width, and 1.29 mm in thickness, made of SAE1070 material, as shown in Figure 1. The mechanical properties of SAE1070 material were considered in the analysis model, as shown in Table 1.

In this study, the Johnson–Cook model, as shown in Equation (1), was incorporated to simulate the deformation behavior that occurs rapidly due to plastic deformation when the shot impacts the material surface. This is because the material experiences a high strain rate during the impact process. The Johnson–Cook model offers the advantages of easy simulation of deformation behavior and simple structure, making it computationally efficient [34].
(1)$$\sigma =[A+B(\varepsilon^{p}{)}^{n}]\left[1+Cln\left(\frac{{\dot{\varepsilon }}^{p}}{{\dot{\varepsilon }}^{0}}\right)\right]\left[1-{\left(\frac{T-T_{r}}{T_{m}-T_{r}}\right)}^{m}\right]$$

The Johnson–Cook model includes several parameters: *A* represents the initial yield stress, *B* is the strain hardening coefficient, *n* is the strain hardening exponent, *C* is the strain rate sensitivity coefficient, $\varepsilon^{p}$ is equivalent plastic strain, and *m* is the thermal softening exponent. ${\dot{\varepsilon }}^{p}$ is the equivalent plastic strain rate, ${\dot{\varepsilon }}^{0}$ is the initial equivalent plastic strain rate, *T* is working temperature, *T_r_* is the room temperature, and *T_m_* is the melting temperature. The values for these parameters are listed in Table 1.

To reduce the analysis time, a quarter-symmetric Almen strip analysis model, as shown in Figure 2a, was used. The boundary conditions for the model were as follows: the endpoints were restrained in the x, y, and z directions (Ux = Uy = Uz = 0), and displacement was constrained in the direction perpendicular to the symmetry plane (Ux = 0, Uy = 0) [32]. To validate the quarter-symmetric analysis model, a full analysis model was also established and analyzed for comparison. The full analysis model, depicted in Figure 2b, had all four endpoints restrained in the x, y, and z directions (Ux = Uy = Uz = 0).

Figure 3 illustrates the phenomenon of random impact of shot balls in the Almen strip analysis model. To simulate the random impact of shot balls, a probability density function was employed using Abaqus software. The shot balls were assumed to be rigid bodies with a diameter of 0.8 mm and a density of 7800 kg/m^3^. Furthermore, it was assumed that the shot balls undergo no deformation during the impacts.

Figure 4a shows the deformation of the 1/4 symmetry model and the full model of the Almen strip after multiple impacts. Figure 4b displays the deformation in the z-direction at points P1 to P5 along the length (y-direction) of the Almen strip, as indicated in Figure 4. P1 represents the endpoint in the length direction (y-direction) of the Almen strip, while P5 represents the central region along the length direction (y-direction) of the strip. The shot ball’s impact velocity was set at 60 m/s, and the impact time was 0.3 ms.

After analysis, it was confirmed that the deformation obtained from the 1/4 symmetric model closely approximated the deformation obtained from the full model in Figure 5. Furthermore, the analysis time required for the 1/4 symmetric model was 82 min, while the analysis time for the full model was 475 min, resulting in a reduction of 393 min, or approximately 82%, in analysis time using the 1/4 symmetric model. Based on these findings, the 1/4 symmetric model was used as the multi-random impact analysis model in this study.

The minimum element sizes are 0.09 mm, 0.08 mm, 0.07 mm, 0.06 mm, and so on. These element sizes were used in Figure 6 to examine the convergence of the deformation measured at the center position of the Almen strip analysis model. In Figure 7, the average deformation obtained after analysis was approximately 1.314 mm, and all four element sizes provided values that were close to the average deformation. Considering the reduction in analysis time, a minimum element size (*Le*) of 0.09 mm was chosen.

Figure 8 shows the deformations of the Almen strip in a 1/4 symmetric analysis model with a minimum element size (*Le*) of 0.09 mm after multiple shot balls impact randomly with velocities of 40 m/s, 50 m/s, 60 m/s, and 70 m/s for a duration of 0.3 ms. From Figure 8, it can be observed that the maximum deformations are 0.674 mm at an impact velocity of 40 m/s, 0.907 mm at 50 m/s, 1.306 mm at 60 m/s, and 1.407 mm at 70 m/s.

Table 2 presents the arc heights obtained from the Almen strip analysis model as a function of the analysis time. These analytical arc heights are calculated based on the maximum deformation of the Almen strip obtained from the analysis, following the experimental measurement method for determining arc height. Specifically, the height of the deflection within the 31.75 mm gauge length of the Almen gauge is calculated to represent the analytical arc height. These analytical arc heights are compared with the experimental arc heights to validate the effectiveness of the proposed discrete-finite element multiple-random shot peening impact analysis in this study. The analysis times are set as 0.3 ms, 0.6 ms, 0.9 ms, and 1.2 ms, with corresponding impact velocities of 40 m/s, 50 m/s, 60 m/s, and 70 m/s for each analysis time.

### 2.2. Experimental Effectiveness of the Almen Intensity Simulation

In order to validate the accuracy of the previously performed discrete-finite element (DE-FE) analysis based on the Almen intensity analysis model, an Almen intensity measurement experiment was conducted in this section. An impeller-type shot peening equipment was used in the experiment as shown in Figure 9, with a diameter of 250 mm. The rotation speed of the impeller was adjusted to achieve shot velocities ranging from 40 m/s to 70 m/s. The Almen strips used in the experiment were of type A, identical to the ones used in the analysis model. The shot used was a round-cut wire shot with an average diameter of 0.8 mm. The shot peening duration was set to 60 s, 120 s, 180 s, and 240 s, with a projection rate of 42 kg/min.

Figure 10 illustrates the process of measuring the deflection height of the curved Almen strip using the Almen gauge after shot peening. The Almen intensity is measured by placing the Almen strip on the upper surface of the Almen gauge and using a dial gauge to measure the deflection height within the 31.75 mm span (L) of the gauge [32,35].

Table 3 presents the measured values of Almen intensity for the Almen strip. It can be observed that the Almen intensity increases with the shot peening time for each impact velocity, and it also increases with the increasing impact velocity for the same peening time. This can be attributed to the fact that an increase in impact velocity results in higher kinetic energy, and an increase in peening time leads to the accumulation of kinetic energy, both of which contribute to the deformation of the Almen strip in proportion.

By comparing the analytical Almen intensity values with the experimental Almen intensity values, we can validate the validity of the analysis technique. Figure 11 shows the comparison between the analytical and experimental Almen intensities. The analytical Almen intensity curve closely matches the experimental Almen intensity curve. For all impact velocities, the analysis time of 0.3 ms corresponds well to the experimental peening time of 60 s, 0.6 ms corresponds to 120 s, 0.9 ms corresponds to 180 s, and 1.2 ms corresponds to 240 s. This means that the analytical Almen intensity curves can be used as a substitute for the experimentally obtained Almen intensity curves, which are essential for actual peening processes. These analytical Almen intensity curves are employed for determining the Almen intensity and residual stress prediction in the shot peening process of real metal materials, as well as in the unit cell analysis models.

## 3. Analytical Peening Residual Stress and Its Experimental Verification

### 3.1. Peening Residual Stress by the DE-FE Analysis

Figure 12 represents a DE-FE multi-random impact unit cell analysis model used to predict residual stresses induced in the metal material after actual shot peening processes. The model is capable of representing the entire surface area. The material used in the model is AISI4340, with dimensions of 1 mm × 1 mm × 3 mm. The same constant probability density function and Johnson–Cook model, as introduced in Section 2, were applied to the Almen strip analysis models. The shot ball model reflects a diameter of 0.8 mm and a density of 7800 kg/m^3^, assuming it is an undeformed rigid body. These shot balls impact at a 90° angle, with impact velocities of 40 m/s, 50 m/s, 60 m/s, and 70 m/s. Contact or impacts between the shot balls are not considered. Table 4 presents the mechanical properties and constant values for the AISI 4340 material and the shot balls.

During the shot peening process, the continuous impacts of shot balls create surface indentations, and significant deformation occurs beneath the impact area. To analyze the deformation of the elasto-plastic specimen, the NLGEOM (non-linear Geometry) option in the ABAQUS Explicit code was utilized. Additionally, a three-dimensional 8-node reduced integration element (C3D8R) was used for meshing [35,36,37]. As for the boundary conditions, the bottom surface of the unit cell was fully constrained (Ux = Uy = Uz = 0), and the four side surfaces were restrained in the direction perpendicular to the surface (Ux = 0, Uz = 0). To incorporate the penalty algorithm for contact, contact surface elements were placed on both sides of the material and shot balls. The analysis involved approximately 12,650 elements and 13,968 nodes. The compressive residual stresses generated by shot peening are primarily embedded in the surface and subsurface regions. Therefore, the finite element mesh was refined in the areas with significant plastic deformation on the surface and coarser in the subsurface regions with smaller plastic deformation. The minimum element size (*Le*) adopted for the analysis was 0.02 mm. To evaluate the residual stresses from the proposed analytical model in this study, an analytical residual stress evaluation method based on XRD experiments, commonly used for measuring residual stresses, is required. Generally, XRD residual stresses are obtained as averaged stresses over a certain area where X-rays are probed. Taking this into consideration, in Figure 12, the evaluation regions were defined into three groups with different numbers of nodes to obtain the analytical average solutions that closely match the experimental field [38]. These evaluation regions consist of 3, 9, and 25 nodes, respectively, and are centered around the analytical model. The evaluation region composed of three nodes is formed by consecutive nodes aligned with the direction of residual stress measurement. The evaluation region composed of nine nodes is a 3 × 3 area, horizontally and vertically, resulting in a total of nine nodes. The evaluation region composed of 25 nodes is a 5 × 5 area, horizontally and vertically, representing the entire set of nodes.

### 3.2. Experimental Verification of Peening Residual Stress

In order to validate the effectiveness of the proposed unit cell analysis model shown in Figure 12, an analysis of multiple random peening residual stresses was performed, considering the actual measured Almen intensity. The analysis considered the average over the evaluation regions, which refers to the multi-node average for the three evaluation regions shown in Figure 12. After the analysis, the obtained peening residual stresses were compared with the experimentally measured residual stresses by XRD in the depth direction from the surface of specimens peened with four different Almen intensities [39] in Figure 13. In case (a), the Almen intensity is 0.0027 inchA (=0.06858 mmA), and the impact velocity is 40 m/s, with an analytical peening time of 0.13188 ms. After the analysis, it can be observed that both the 9-point and 25-point averaged solutions are closer to the XRD experimental residual stress than the 3-point one. Case (a) represents the scenario where the maximum compressive residual stress occurs at the surface. Considering both the surface and maximum compressive residual stress, the 25-point average solution is slightly closer to the XRD experimental results than the 9-point average solution. In case (b), the Almen intensity is 0.0063 inchA (=0.16002 mmA), and the impact velocity is 50 m/s, with an analytical peening time of 0.25949 ms. After the analysis, it is observed that the 25-point averaged solution is closer to the XRD experimental residual stress than the 3-point and 9-point averaged solutions. In case (c), the Almen intensity is 0.0083 inchA (=0.21082 mmA), and the impact velocity is 60 m/s, with an analytical peening time of 0.25502 ms. After the analysis, it is found that the 3-point, 9-point, and 25-point averaged solutions are close to the XRD experimental residual stress. In case (d), the Almen intensity is 0.0141 inchA (=0.35814 mmA), and the impact velocity is 70 m/s, with an analytical peening time of 0.60531 ms. After the analysis, it is observed that both the 9-point and 25-point averaged solutions are close to the XRD experimental residual stress, but the 25-point average solution is closer to the XRD experimental residual stress at different depths. Overall, based on the comparison with XRD experimental residual stress [39], the 25-point averaged solution is found to be more effective in all cases. Based on the results shown in Figure 13, the discrete-finite element model that incorporates the actual Almen intensity and provides the 25-point averaged solution is chosen as the optimal peening analysis model.

The effectiveness of shot peening is typically evaluated based on the magnitude of surface residual stress and maximum compressive residual stress. Table 5 presents the magnitudes of analytical and experimental SRCS, along with the corresponding errors. The error ranges from 0.75% to 6.55%, indicating a good agreement between them. Additionally, Table 6 shows the magnitudes of analytical and experimental MRCS, along with the corresponding errors. The error ranges from 1.58% to 7.93%, demonstrating an even better agreement compared to the surface residual stress. The reason for the larger error in surface residual stress compared to maximum compressive residual stress is attributed to the sensitivity of the experimental measurement positions and directions to the irregular shape of surface indentation generated by randomly impacting shot-peening balls on the material surface. In addition, it is believed that the analysis model’s surface grid, which constitutes the surface region of the model, undergoes irregular deformation due to the random impacts of shot peening balls, resulting in a greater stress interference effect compared to the interior grid of the analysis model. Figure 14 illustrates the distribution of residual stress within the analysis model after shot peening analysis. It shows a trend where the compressive residual stress increases and then decreases as the depth from the surface increases. This trend is consistent with the distribution of measured residual stress typically observed after conventional shot peening processes.

## 4. Conclusions

In this study, a new combined discrete-finite element analysis model incorporating both discrete elements and finite elements was used to simulate the residual stress distribution in actual shot peening processes. The conclusion is as follows.

A discrete-finite element analysis model was established for the Almen strips used in measuring Almen intensity, allowing us to obtain the deformation quantities via the analysis of multiple random impacts. The discrete element analysis simulated the random behavior of numerous shot-peening balls, while the finite element analysis verified the deformation behavior of the Almen strips. It was also observed that the deformation increased proportionally with the increasing impact velocity of the shot balls. By converting these deformations into Arc heights, the analytical Almen curve was obtained. Additionally, the experimental Almen curve based on the actual peening time was measured and matched with the analytical curve. The results showed a good agreement between the analytical and experimental Almen curves for different peening times. This confirmed that the analytical Almen curve could effectively replace the experimental Almen curve based on the peening time. Afterward, the analysis model was expanded to predict the actual residual stresses in shot peening. This analysis model was constructed using unit cells, and an analysis technique was applied where the stresses obtained at multiple points within the effective area were averaged to evaluate the area-averaged stresses. In this study, the analysis was performed on AISI 4340 material that was shot peened under four different Almen intensities, obtaining the residual stresses. Then, these analysis results were compared with the experimental residual stresses obtained from specimens shot peened with the same Almen intensities. The results showed that the surface residual stresses with an error range from 0.75% to 6.55% and the maximum compressive residual stresses with an error range from 1.58% to 7.93% obtained from the analysis were in close agreement with the experimental residual stresses.

Ultimately, the proposed discrete-finite element random impact analysis model, which incorporates Almen intensity-dependent arc height, has been validated for its effectiveness and validity as an analysis technique that can replace actual shot peening processes. This analysis technique can be effectively used in various industries to apply durable mechanical parts with optimal peening intensity.

## Figures and Tables

**Figure 1 materials-16-05472-f001:**
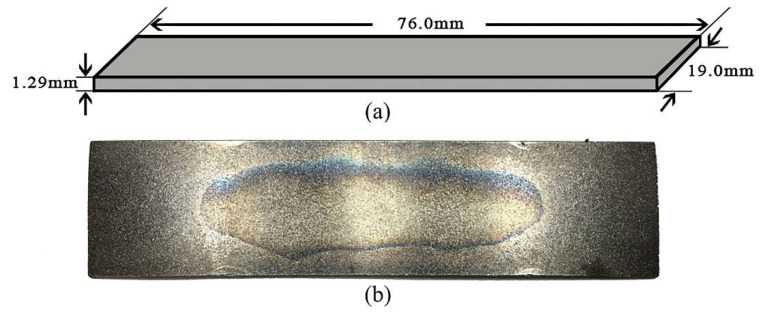
Sample. (**a**) geometric dimensions; (**b**) an Almen strip of type A.

**Figure 2 materials-16-05472-f002:**
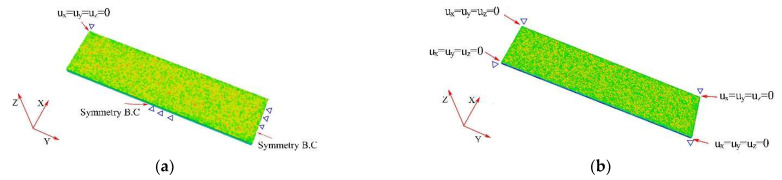
Boundary conditions for the quarter Almen strip model and the whole Almen strip model. (**a**) quarter model; (**b**) full model.

**Figure 3 materials-16-05472-f003:**
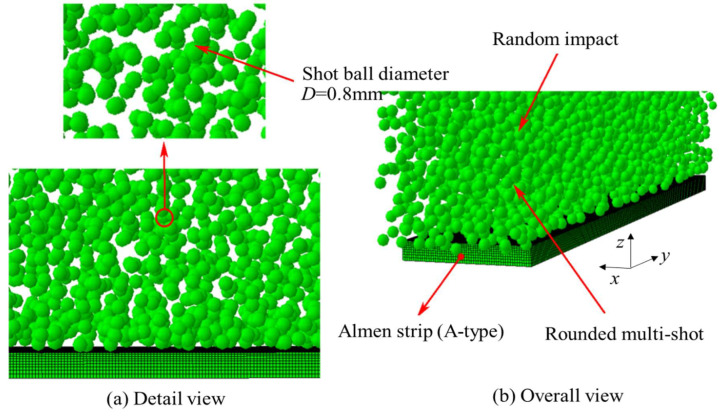
The 3D discrete element analysis model with multi-shot random impact on the surface of the Almen strip. (**a**) detail view; (**b**) overall view.

**Figure 4 materials-16-05472-f004:**
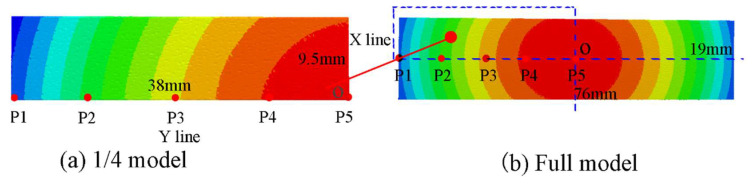
Deflection of the Almen strip. (**a**) deflection of the quarter Almen strip; (**b**) deflection of the full Almen strip.

**Figure 5 materials-16-05472-f005:**
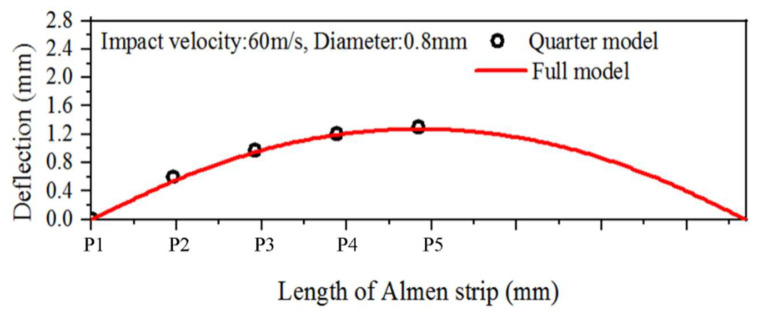
Deflections for the full model and quarter model of the Almen strip.

**Figure 6 materials-16-05472-f006:**
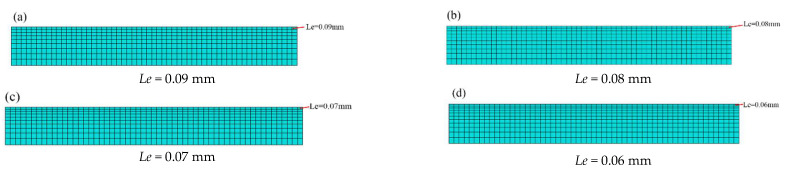
The various element minimum sizes of the Almen strip DE-FE analysis model.

**Figure 7 materials-16-05472-f007:**
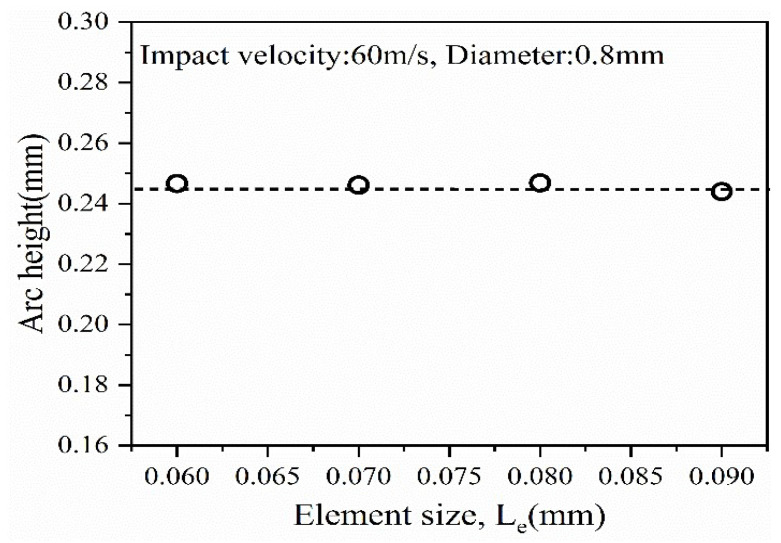
Convergence of arc height with various mesh sizes.

**Figure 8 materials-16-05472-f008:**
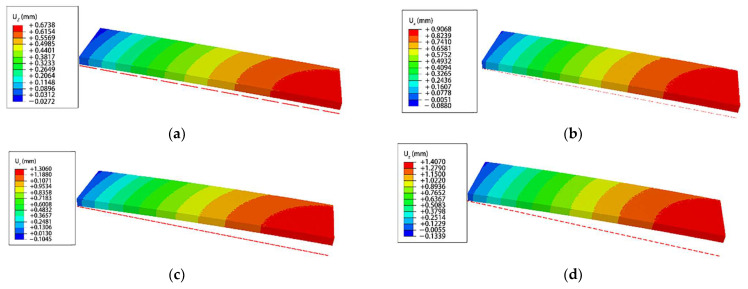
The deformation of the Almen strip in the z-direction, when (**a**) the peening velocity is 40 m/s; (**b**) the peening velocity is 50 m/s; (**c**) the peening velocity is 60 m/s; (**d**) the peening velocity is 70 m/s.

**Figure 9 materials-16-05472-f009:**
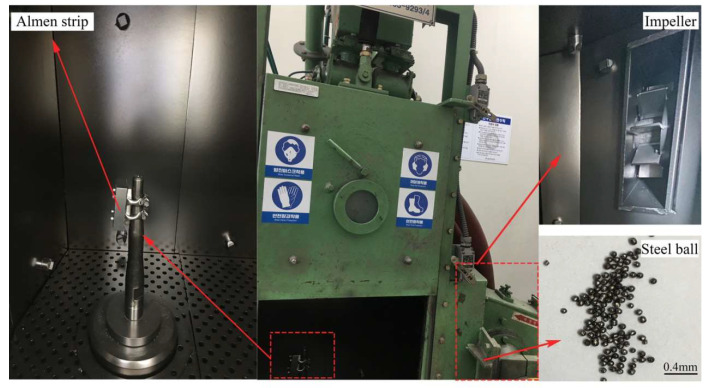
Shot peening equipment.

**Figure 10 materials-16-05472-f010:**
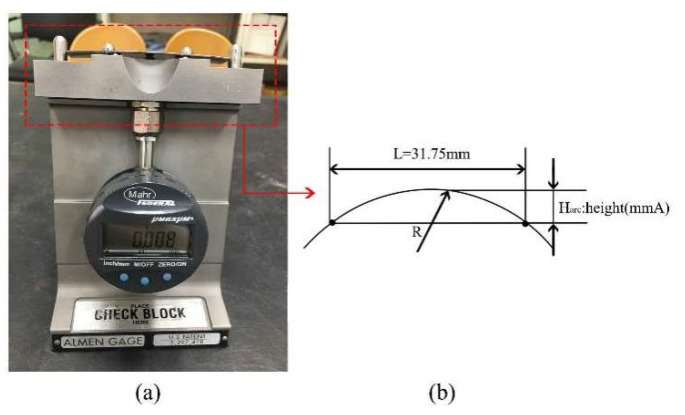
Measurement devices. (**a**) Almen gage; (**b**) Arc height calculation principle.

**Figure 11 materials-16-05472-f011:**
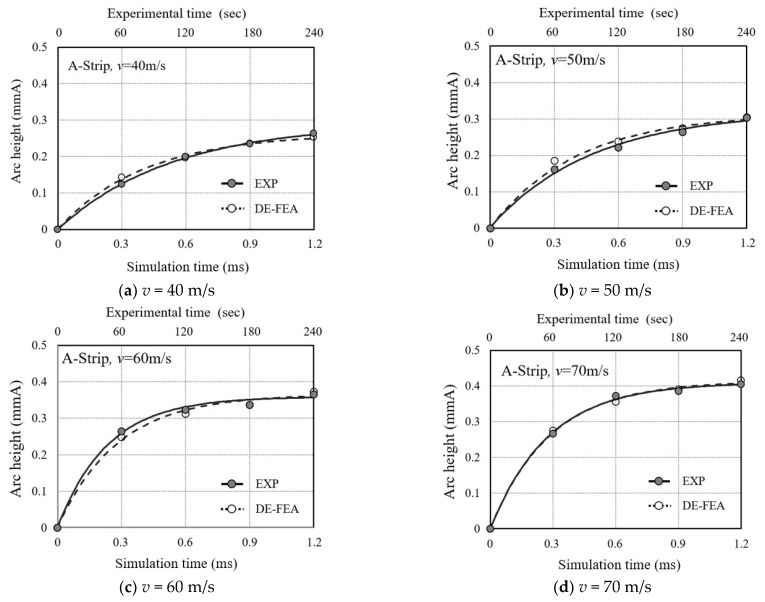
Arc height of the Almen strip at different impact velocities (**a**) with an impact velocity of 40 m/s; (**b**) with an impact velocity of 50 m/s; (**c**) with an impact velocity of 60 m/s; (**d**) with an impact velocity of 70 m/s.

**Figure 12 materials-16-05472-f012:**
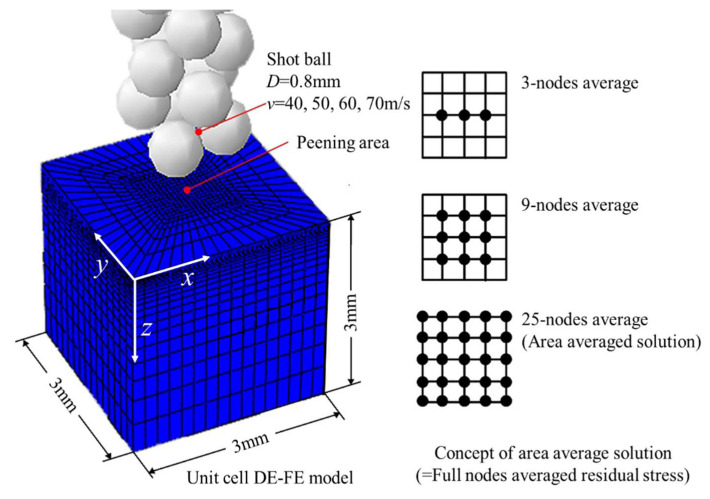
The concept of the area-averaged (=full node averaged) residual stress from unit cell FE model.

**Figure 13 materials-16-05472-f013:**
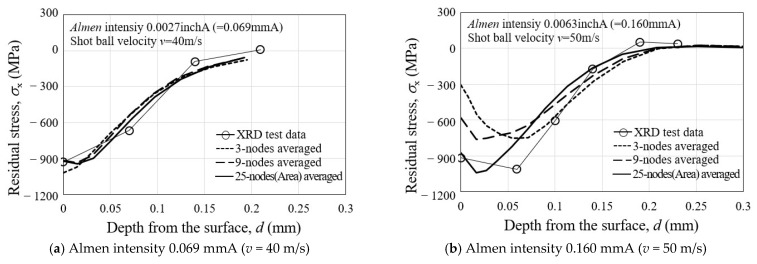
Comparison of the DE-FE analytical residual stress and XRD test residual stress [39] with four different Almen intensities with (**a**) the Almen intensity is 0.069 mmA (*v* = 40 m/s); (**b**) the Almen intensity is 0.160 mmA (*v* = 50 m/s); (**c**) the Almen intensity is 0.211 mmA (*v* = 60 m/s); (**d**) the Almen intensity is 0.358 mmA (*v* = 70 m/s).

**Figure 14 materials-16-05472-f014:**
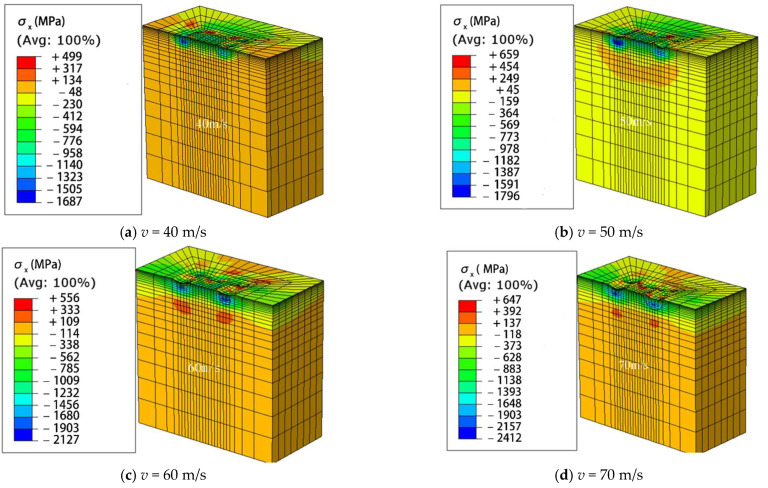
Different surface morphologies at different peening velocities (**a**) with an impact velocity of 40 m/s; (**b**) with an impact velocity of 50 m/s; (**c**) with an impact velocity of 60 m/s; (**d**) with an impact velocity of 70 m/s.

**Table 1 materials-16-05472-t001:** Coefficients of Johnson-Cook model adopted for deformation analysis of Almen strip [10].

Material	*E* (GPa)	Poisson’s Ratio	*ρ* (kg/m^3^)	*A* (MPa)	*B* (MPa)	*n*	*m*	*T*_m_ (K)	*T*_r_ (K)	*C*
SAE 1070	205	0.29	7800	1408	600.8	0.234	1.0	1793	298	0.0134
Shot ball	210	0.30	7800							

**Table 2 materials-16-05472-t002:** Arc height by DE-FE analysis.

Analysis Time *t* (ms)	Arc Height (mmA)			
	40 m/s	50 m/s	60 m/s	70 m/s
0.3 ms	0.124	0.161	0.243	0.266
0.6 ms	0.198	0.221	0.322	0.371
0.9 ms	0.233	0.254	0.335	0.401
1.2 ms	0.243	0.293	0.361	0.438

**Table 3 materials-16-05472-t003:** Arc heights by Almen test.

Exposure Time *t* (s)	Arc Height (mmA)			
	*v* = 40 m/s	*v* = 50 m/s	*v* = 60 m/s	*v* = 70 m/s
60	0.143	0.185	0.246	0.275
120	0.197	0.238	0.312	0.355
180	0.235	0.274	0.336	0.390
240	0.252	0.304	0.372	0.416

**Table 4 materials-16-05472-t004:** Coefficients of Johnson-Cook model adopted for residual stress analysis of the unit cell DE-FE model. [10].

Material	*E* (GPa)	Poisson’s Ratio	*ρ* (kg/m^3^)	*A* (MPa)	*B* (MPa)	*n*	*m*	*T*_m_ (K)	*T*_r_ (K)	*C*
AISI4340	210	0.30	7800	1498	943.8	0.260	1.03	1793	298	0.014
Shot ball	210	0.30	7800							

**Table 5 materials-16-05472-t005:** Comparison of the numerical and experimental results of the surface residual compressive stress (SRCS).

Shot Ball Velocity (m/s)	Arc Height (mmA)	Analysis *σ_src_*_s_/MPa	XRD Test [39] *σ_src_*_s_/MPa	Error (%)
40	0.069	−922.986	−929.961	0.75
50	0.160	−874.086	−917.51	4.73
60	0.211	−911.864	−975.828	6.55
70	0.358	−730.179	−781.25	6.53

**Table 6 materials-16-05472-t006:** Comparison of the numerical and experimental results of the maximum residual compressive stress (MRCS).

Shot Ball Velocity (m/s)	Arc Height (mmA)	Analysis *σ_src_*_s_/MPa	XRD Test [39] *σ_src_*_s_/MPa	Error (%)
40	0.069	−944.677	−929.961	1.58
50	0.160	−1041.03	−1010.89	2.98
60	0.211	−1067.78	−1159.84	7.93
70	0.358	−1211.59	−1256.25	3.55

## Data Availability

Not applicable.

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
