# Peer review of "A Discrete-Finite Element Analysis Model Based on Almen Intensity Test for Evaluation of Real Shot Peening Residual Stress"

_materials, 2023, doi:10.3390/ma16155472_

Round 1

Reviewer 1 Report

Comment 1 The title of the paper, keywords, abstract, and introduction are in accordance with the content of the paper.

Comment 2 Line 50, I quote: "... and Guagliano et al. proposed ...", it is appropriate to mention the name of the first author, in this case, it is "Gangaraj".

Comment 3 It would be appropriate to indicate what is a measurable output of discrete element analysis and what is a measurable output of finite element analysis.

Comment 4 The conclusion should be supplemented with quantitative data on the results achieved.

Author Response

Dear Reviewer 1

We are very grateful to reviewer for reviewing the paper so carefully. According to your nice suggestions, we have made extensive corrections to our previous draft, the detailed corrections are listed below:

Comment 1 The title of the paper, keywords, abstract, and introduction are in accordance with the content of the paper.

Answer: 

(1) The titel was modified base on the main text as follows

“A discrete-finite element analysis model based on Almen intensity test for evaluation of real shot peening residual stress”

(2) The abstract also has been revised to emphasize the purpose of this manuscript.

“In this study, a combined discrete-finite element model based on Almen intensity measurement test was poroposed to evaluate the real shot peening residual stress.”

(3) Additionally, the keywords has been revised as below.

“Keywords : Shot peening, Residual stress, Discrete element analysis, Finite element analysis, Almen intensity”

(4) Finally, the last part of the introducton has been modified to match the contents of the main text.

“In this study, a new analysis technique is proposed to evaluate the real shot peening residual stress using the discrete-finite element analysis model based on Almen intensity measurement test.”

Comment 2 Line 50, I quote: "... and Guagliano et al. proposed ...", it is appropriate to mention the name of the first author, in this case, it is "Gangaraj".

Answer: It has been modified at the 50 lines of the paper.

Comment 3 It would be appropriate to indicate what is a measurable output of discrete element analysis and what is a measurable output of finite element analysis.

Answer: 

At the end of the introduction, the factors obtained after discrete element analysis and the factors obtained after finite element analysis are clearly specified. 

“Here, the discrete element analysis is used for random behavior of numerous shot balls and the impact load is obtained when the shot balls impact on the surface of the model, while finite element analysis is used for deformation behavior of the model and residual stress is indicated.”

Comment 4 The conclusion should be supplemented with quantitative data on the results achieved.

Answer: 

The key point of this paper is that the analysis results for the surface residual stress and the maximum compressive residual stress are close to the experimental results. Therefore, in the conclusion, the error ranges of surface residual stress and maximum compressive residual stress were expressed as quantitative values, respectively. 

“The results showed that the surface residual stresses with error range from 0.9% to 6.4% and the maximum compressive residual stresses with the error range from 0.9% to 2.3% obtained from the analysis were in close agreement with the experimental residual stresses.”

Reviewer 2 Report

Dear Authors,

The paper outline and describe an interesting approach and FEM model for the simulation of the final results in shoot-peening process.

The Almen strip approach has been adopted and it is intersting.

The authors should clarify why they have adopted SAE1070 for obtaining  the analytical Almen intensity and how it is mandatory this material for the AISI 4340 comparison results. In a better way: is the model calibration dependant upon the “starting material” selected (in this case SAE 1070) ?

How  independent is the FEM model with respect to a general steel application.

Please, give some more details about the general application and further set-up tests required for a general material (even the ligth alloys behaviour can be modellised ?)

Moreover,

I am really surprised from the very low errors difference from the simulated and experimental results in  Tab.5. I am not sure to have seen in the laboratory a so beutifull correlation. So please give some more details/comments.

Do you have also considered some direct  residual stress measurement methods  based on XR, i.e ? 

Author Response

Dear Reviewer 2

We are very grateful to reviewer for reviewing the paper so carefully. According to your nice suggestions, we have made extensive corrections to our previous draft, the detailed corrections are listed below:

  1. The authors should clarify why they have adopted SAE1070 for obtaining  the analytical Almen intensity and how it is mandatory this material for the AISI 4340 comparison results. In a better way: is the model calibration dependant upon the “starting material” selected (in this case SAE 1070) ?

Answer :

SAE1070 used in this study is the material of Almenstrip. And Almenstrip is a measurement sample standardized by SAE of the United States, and must be used during the shot peening process. Therefore, it was necessarily used in this study. In addition, AISI4340 is a material widely used when manufacturing gears or landing gears, and the shot peening process is always required. These should be subjected to shot peening treatment with a standardized Almen strength. Therefore, in this study, AISI4340 material and Almen strip material SAE1070 are closely related.

  1. Please, give some more details about the general application and further set-up tests required for a general material (even the lightalloys behaviour can be modelized ?)

Answer:

In the case of Almen strip used in the shot peening process, there are three types This study was conducted on the main metals frequently used in the shot peening process. This paper presents the research results for optimizing the shot peening process of mechanical parts that require durability. Therefore, AISI4340 material, which is a carbon steel material, was adopted. In addition, the analysis method proposed in this study is the same for aluminum alloy and titanium alloy, which are lightweight materials for aviation. If the Almen strength is weak, use a thin “N” type Almen strip. In this case, since the deformation amount can be increased, the deformation behavior of the light alloy can be predicted easily and accurately. Recently, Almen strips made of aluminum have been developed, which are used to measure the Almen strength of non-ferrous metals. As such, this study proposes a discrete-finite element analysis method based on the Almen intensity. Therefore, it is sufficiently used to predict the deformation behavior and residual stress of various metal materials.

  1. Iam really surprised from the very low errors difference from the simulated and experimental results in  Tab.5. I am not sure to have seen in the laboratory a so beutifull correlation. So please give some more details/comments.

Do you have also considered some direct  residual stress measurement methods  based on XR, i.e ? 

Answer:

The methods to verify the shot peening effect are the size(=magnitude) of the surface residual stress, the maximum compressive residual stress, and the thickness of the compressive residual stress layer. The surface residual stress has the effect of delaying the initiation period of cracks caused by tensile fatigue load, and the maximum compressive residual stress has the effect of slowing down the propagation speed of cracks. Therefore, in this study, the error just was calculated by comparing the analysis results and the experimental results for the size of the surface residual stress and the maximum compressive residual stress. In Figure 13, the XRD experimental values ​​show the data points of 4 to 6 measurements. That is, actual experimental values ​​are not continuous measurement results. Therefore, after grasping the trend of the experimental values ​​with numerical graphs, they were compared with the analyzed values. Also, the calculated error is not a comparison of stresses obtained at the same depth. Therefore, the error is slightly larger at the same depth. In this study, it is a calculated value with an emphasis on the fact that the size of the surface and maximum compressive residual stress is important. In addition, the actual experimental surface residual stress value is measured after electrolytically polishing the surface. However, the analysis surface residual stress in this study is the value obtained from the analysis model surface. Therefore, if the depth difference is considered, the error may also increase. This will be investigated through follow-up studies.

Please refer to the attached file for additional answers.
